# Modeling of Cowpea (*Vigna unguiculata*) Yield and Control Insecticide Exposure in a Semi-Arid Region

**DOI:** 10.3390/plants10061074

**Published:** 2021-05-27

**Authors:** Messias de Carvalho, Wiktor Halecki

**Affiliations:** 1Department of Agronomy, Kimpa Vita University, Henriques Freitas Street 1, Uíge 77, Angola; messias.monteiro.carvalho@student.urk.edu.pl; 2Department of Soil Science and Agrophysics, Faculty of Agriculture and Economics, University of Agriculture in Krakow, Al. Mickiewicza 21, 31-120 Krakow, Poland; 3Department of Hydrology, Meteorology and Water Management, Warsaw University of Life Sciences, Nowoursynowska Street 166, 02-787 Warsaw, Poland

**Keywords:** local genotypes, multivariate analysis, phytosanitary control, *Vigna unguiculata* L. Walp., resistant plant

## Abstract

The aim of this study was to evaluate the adaptability of different genotypes of cowpea (*Vigna unguiculata* L. Walp.) in the edaphoclimatic conditions of a semi-arid region. In the experimental design, a completely randomized split-plot (2 × 8), with 3 repetitions (blocks) was used. The experiment comprised 7 new genotypes and 1 local genotype as the first main factor and application of insecticide as a secondary factor. Two-factor analysis of variance (two-way ANOVA) determined the differences between the treated and untreated plots. The results obtained in the experiment showed that the introduced genotypes V3 (IT07K-181-55), V7 (H4), and V5 (IT97K-556-4M) adapted well to the edaphoclimatic conditions of the study area and their yields were respectively 1019, 1015, and 841 kg/ha of grains in treated plots and 278, 517 and 383 kg/ha in untreated plots. Multivariate analysis revealed that the most important parameter was the germination rate. Finally, the best yield was obtained with the genotype V3 (IT07K-181-55), subjected to the use of insecticide, and with the V7 (H4) genotype in untreated plants. The findings presented in this research should be useful in crop system agricultural programs, particularly in the terms of selection of cultivating systems suitable for high-yielding cowpea.

## 1. Introduction

Agricultural plants play an important role in the nutritional balance in semi-arid areas [1,2,3,4,5,6,7,8,9,10]. Currently, many researchers have focused a lot of effort into improving yields and increasing the production of crops [7,11,12,13]. Cowpea (*Vigna unguiculata* L. Walp.) is an extensively cultivated plant, widely used as a legume in native tropical and subtropical zones or the semi-arid regions of South Africa [14]. *Vigna unguiculata* in rural areas (known as “Macunde beans”) is a source of green manure, organic matter [15] for hay, forage and silage [16]. The goal of minor farming practices is to improve a cultivation system that is highly adaptable to changes in the soil condition [17,18]. Studies have documented that *V. unguiculata* can grow in agroforestry and orchards as an important protein plant to the human diet [19]. Other research showed a valuable improvement in yields [20,21,22,23]. In agroecological zone of Cameroonian Guinea, Sudano, and the Sahelian savanna, *V. unguiculata* is often combined with sorghum [24]. Therefore, it is one of the protein-rich plants able to combat food insecurity, malnutrition and poverty in sub-Saharan African populations [25,26]. Furthermore, many factors limit crop production, including biological ones [27,28,29,30]. Moreover, the low production of local genotypes, lack of improved genotypes and the use of traditional agricultural techniques in arid farming conditions strongly decrease the crop potential to grow [28,31,32,33,34]. In the western and central regions of Africa, *V. unguiculata* is used as a suitable plant to balance grain and forage nutrition in the post-harvest period [35].

Variegated grasshopper—*Zonocerus variegatus* (L.) (Orthoptera: Pyrgomorphidae) is a polyphagous insect that feeds on a wide range of cultivated and uncultivated land [36]. Reports of damage cowpea by *Zonocerus variegatus* has increased in recent years [37]. In the agricultural production systems of the humid forest zone, activity of this pest results in the defoliation of leaves and total damage to food crops [38]. Agricultural production in Angola cannot ensure food security from the socio-economic point of view. Thus, farmers are at risk of starvation [39]. Hence, the goal of an alternative farming system is to help increase the availability [40,41,42], affordability [43,44,45,46,47,48,49,50] of crops and meet their nutritional needs [51,52,53,54,55,56,57,58,59,60]. In Angola, *V. unguiculata* belongs to the important staple crop. Therefore, we chose this crop as an example of a nutritious, protein-rich, soil-tolerant and largely tasty food source. The main aim of the study was to assess the adaptability of various genotypes of *V. unguiculata* to edaphoclimatic conditions. We hypothesized that the newly introduced and selected genotypes of *V. unguiculata* may be more productive than the local genotypes. Consequently, our objectives were: (i) to select the best genotypes for the insecticide-exposed crop system; (ii) evaluating of the performance of drought-prone genotypes in the Sub-Saharan region; (iii) highlighting selected genotypes that can be sufficiently adapted to semi-arid conditions; (iv) assessment of the main factors influencing the seed and grain yield using multivariate analysis.

Currently, there is no data on a legume-food product in Angola. The article presents the first application of the cultivation system in the Uíge province to maintain high agricultural productivity credited in the scientific literature.

## 2. Results

### 2.1. Grain Production

Highly significant differences (*p* < 0.001) were observed both between genotypes and between treatments with insecticide and without insecticide (Table 1). Mean values of grain yield of the insecticide use on grain yield of different genotypes of *V. unguiculata* are presented in Figure 1. Grain yield of the genotypes ranged from 307 to 1019 kg/ha with the application of insecticide and 113 to 517 kg/ha without treatment with an insecticide. Thus, in relation to grain production, three genotypes were highlighted: IT07K-187-55 (V3), H4 (V7), and IT97K-556-4M (V5). These produced respectively 1019, 1015, and 841 kg/ha of grain with the application of insecticide and 278, 517, and 383 kg/ha of grain without application of insecticide. Canonical Variety Analysis (CVA) plotted to graph detected relation between genotypes according to grain yield. CVA led to enhanced detection of grain patterns of variation across all analyzed groups. 68% of the total variation was accounted for the first two canonical axes (Figure 2).

### 2.2. Seed Production

Seed yields ranged widely, from 293 to 992 kg/ha of seeds with insecticide application and 103 to 501 kg/ha of seeds in genotypes untreated with insecticide. In fact, these genotypes were: IT07K-187-55 (V3), H4 (V7), and IT97K-556-4M (V5) which respectively produced 992, 991, and 811 kg/ha of seeds with insecticide application and 251, 501, and 364 kg/ha of seeds untreated with insecticide. The effect of insecticide on seed yield of different genotypes of *V. unguiculata* is presented in Figure 3. In the case of grain production, highly significant differences were also observed both among genotypes (*p* = 0.005) as well as among treated/untreated with insecticide (*p* = 0.0027), with respect to seed yield. Principal component axes PCA 1 (40.81%) and PCA 2 (21.85%) explained the variation (Figure 4) of crop parameters revealing loading factors (Table 2) and eigenvectors (Table 3). Germination rate was ranked as the most important across all the studied plant traits (Figure 5). In the present study, the correlation analysis showed that the measured parameters affected plant growth (Table 4). Furthermore, a generalized additive model (GAM) described that seed yield may be a qualitative variable used as indicator of *Zonocerus variegates* occurrence (Table 5, Figure 6a,c). Additionally, grain yield, exposure of disease, the weight of selected seeds, and insect risk have determined the main parameters affecting genotypes (Table 6, Figure 6b,d).

## 3. Discussion

### 3.1. Influence of Pesticides on Germination and Anthesis

Damage caused by pests can reach up to 80–100% in the case of agricultural treatment [61,62,63,64] for effective plant management [65,66,67,68,69,70]. The results of this study showed that we detected a highly significant difference between the genotypes in relation to the percentage of germination (Table 1). In general, the genotypes IT07K-311-1 (V4), IT04K-221-1 (V1), DIAMANTE (V6), IT07K-187-55 (V3), and IT89KD-288 (V2) gave the highest significant rate, while genotypes IT97K-556-4M (V5) and H4 (V7) presented a below-average germination rate. The findings of this study lodged that the application of insecticide did not influence the number of days until flowering, the number of gaining maturity days, and the weight of 100 seeds. However, significant differences were observed between the genotypes in relation to the number of days needed until ripeness. The number of days until flowering (44–53 days, the average of days was 50), and the number of days to maturity (72–82 days, the average of days was 75), as well as the weight of 100 seeds (10–18.1 g), were linked to the genotype. 

Data presented in Table 4 revealed that plants untreated with insecticide exhibited a decrease in yield when compared to those treated with insecticide. Nevertheless, results indicated that there were significant differences between the genotypes for the risk of disease (*p* < 0.05). The results showed that there was no significant difference when comparing the treatment with insecticide and without regard to the number of days until matureness (Table 6). Therefore, the average number of pods per plant was in the range of 8–12 per plant and was entirely different from data (6.0–52.0 pods per plant) published [71]. Thus, among the genotypes, the number of days before maturity varied between 72–82 days after sowing. Meanwhile, the local insecticide-treated genotype was able to grow with no signs of pest damage and virus disease. It should be strongly noted that the untreated genotype was susceptible to attacks by *Zonocerus variegatus.*

### 3.2. Selection of the Best Genotypes for the Field

In general, independently of insecticide application, the highest grain and seed yields of *V. unguiculata* were obtained with the genotypes IT07K-187-55 (V3), H4 (V7), and IT97K-556-4M (V5). Data recorded from sowing to harvest showed that all studied genotypes have a middle biological cycle (Figure 4). In addition, there were significant differences between treated and untreated genotypes according to yields of seeds and grains. The application of insecticide contributed to an increase of yield of 154.14% of grains (684 kg/ha) compared to the treatment without application of insecticide (261 kg/ha). Moreover, insecticide use also influenced the increase in seed yield of 173.78% (616 kg/ha), followed by those subjected to untreated with insecticide (225 kg/ha). The increase in grain and seed production may be explained by one factor controlling the impact of insects on insecticide application before and after flowering (Figure 5), and therefore improved the average number of pods per plant. The results revealed that different genotypes of plants treated with insecticide, produced on average 10.7 pods per plant, whereas, untreated with insecticide 8.4 pods per plant. This research is in line with those of [72], who observed that pests are the main constraints to the production of *V. unguiculata* in many parts of Africa. The effect of plant yield was analyzed using the Generalized additive models (GAM). The following parameters: the threshing yield, the weight of seeds selected, the germination rate, the weight of 100 grain, the yield (grain), the number of harvested plants and the number of pods per plant, proved that seed yield may be considered as a biomass indicator and have an impact on the growth (Table 5). Our results showed the increase of germination rate (Figure 6a), and central lines (Figure 6b). Additionally, we reported number of harvested plants since abundance of *Z. variegatus* in short fallows and adjacent fields was no observed (Figure 6c), and the decrease for number of live plants in the border (Figure 6d). In addition, the DCA exhibited that qualitative variables were the germination rate, the weight of 100 grain, the threshing yield, and the grain yield. Thus, selection based on these traits could be applied in this semi-arid area. 

### 3.3. Control Management of Insecticide in Semi-Arid Region

The total seed yield with and without insecticide was 6.338 kg/ha and this seed yield was lower than that related by [73], where 6.732 kg/ha was obtained. The average grain yield (475 kg/ha) and seed (452 kg/ha) were higher than the average yield of 300–400 kg/ha as reported [73]. However, these average yields were different in relation to research proposed [74,75] where average yields were of 724–844 kg/ha, and 526.5–7645.1 kg/ha. The average seed yield treated with insecticide (654.3 kg/ha) and untreated with insecticide (266.1 kg/ha) in our study was lower than the value achieved [76], reaching 60.7–1184.2 kg/ha. Conversely, the yields obtained in grains (684 kg/ha) were lower than the potential yield (1500 kg/ha of grains) with insecticides used for genotypes recorded by the National Institute of Research and Agronomic Studies (INERA) [77]. The low yield achieved was possibly influenced by edaphoclimatic factors: an abundance of rainfall during the experimental period, nutrient insufficiency, soil texture (sandy clay), and soil acidity slightly elevated in relation to that required by the *V. unguiculata*. The best yields of the *V. unguiculata* are obtained in drained sandy soils, little drained clay soils with a pH ranging between 6 and 7 [78]. 

Food security in the Republic of Angola is based mainly on agricultural production. In the Dange-Quitexe municipality (Province of Uíge), the effective increase in the production of *V. unguiculata* should be considered with decisive techniques in the cultivation of this crop in the subsequent experimental agricultural system [79]. When used as a groundcover, *V. unguiculata* constitutes in semiarid and arid region most widely covered by native legume plant and Africa’s basic nutrition. In the semi-arid region, we recommend this plant as a grain legume, vegetable, and fodder crop. 

## 4. Materials and Methods

### 4.1. Study Area

The study was carried out in the Municipality of Dange-Quitexe—Cahunda Village, Uíge Province, Angola in a rainy tropical climate and dry season. The study area (φ = 15°07′05″ and λ = 8°04′05″) is occupied by herbaceous vegetation, predominated by: *Imperata cylindica*, *Smilax acepts*, *Pteridium aquilinum ssp. africanum*, *Sarcocephalus latifolius and Chromoleuma odorata*. The parcel of land is slightly irregular, along Mufula and Loma rivers. 

The study area is characterized by sandy loam soil, with pH ranging from 5–6. The average annual temperature in the rainy season is 23 °C and 21 °C in the dry season. In the rainy season air humidity reaches 85–90%, while in the dry season it is 70–75% [79]. The lowest annual average precipitation is 100 mm and the highest is 1750 mm. The experiment was carried out during the rainy season within first agricultural season in Angola from September to February (Table 7).

### 4.2. Sampling Method Experimental Layout and Design

The seven genotypes (DIAMANTE, H4, IT04K-221-1, IT0K-187-55, IT07K-311-1, IT89KD-288 and IT97K-56-4M) were obtained from the Institut National pour l’Etude et la Recherche Agronomiques (INERA) 13, Av de Cliniques c/Gombe KINSHASA Répulique Démocratique du Congo. The crop was established in local agro-pastoralist perspectives for sustainable production proposed by [77]. We used 7—new genotypes + 1 local genotype. Seeds were sown 5 to 7 cm deep with 18 cm row spacing. Each plot comprised rows of 2.25 m long and 3 m wide, and a spacing of 1.25 m width between rows and 28.5 m between blocks (Figure 7). Insecticide were applied against *Z. variegatus* [80] to each plot at the rate of 20 mL per 16 L (water) in the form of an aerosol, which was pipetted at planting and drifted into plants (two times per week). Selective substance—cypermethrin (25%) was applied [81]. The total area of the field was 826.5 m^2^. The data collection area comprised two central crop-systems, with 0.5 m of each end of the sub-plots and the two lateral lines used as borders. The data collected in the area of each field plot was based on 64 plants randomly extracted from the total population of 128 plants. The experiment was performed according to a randomized complete block design presented as a split-plot (2 × 8), with 3 replicates (blocks). In this cropping system, we incorporated two fundamental factors, namely genotype, as the main factor and insecticide (cypermethrin) use as the second factor. Each block was composed of two plots subdivided into 16 subplots, treated (eight) and untreated (eight) with insecticide that were denominated as T1 (treatment without the use of insecticide), and T2 (treatment with the use of insecticide).

Phenological characteristics related to the plant yield as well as seed production in agriculture system [82] and pesticide interaction were detected [83]. The aphid-mosaic virus in seed was detected by a growing-out test. Risk of disease has been evaluated on a 1–9 scale (>7 severity of insect risk). Phytosanitary control was reported according to the rating scale:

1 = no attack

2–3 = slightly

4–6 = moderate

7–9 = severe

Plants were harvested to estimate seed and grain yield per plot, which was subsequently converted to tonne per hectare (t ha^−1^). Meanwhile, more experimental details are showed in Figure 2. Only African variegated grasshopper *Zonocerus variegatus* (L.) (Orthoptera: *Pyrgomorphidae*) was simultaneously present in the experimental design. Germination rate was obtained by the number of germinated seeds. Flowering was determined according to the number of days from sowing until flowering. Maturity was determined based on the number of days from sowing until maturation. A number of pods per plant (NPP) were obtained by the average number of pods collected at random from 10 plants. Grain yield (GY) was determined by the amount of grains not selected which means combination of low and high quality. Seed yield (SY) was determined by the number of selected grains, which presented better quality (bigger size, vigorous and higher weight)**.** Weight of 100 grains (WHG) was obtained by the means of the total weight of seeds per treatment. Total yield (TY) was determined as a function of the total productivity of the studied area.

### 4.3. Statistical Data Handling

One-way analysis of variance (ANOVA) has proven as an excellent technique to summarize and determine the differences between the genotypes treated (eight) and untreated (eight) with insecticide, using Statistical Software 8.0. In this study, a total of 7 genotypes were characterized. PCA (Principal component analysis) was a useful technique to emphasize the most important plant parameters. Group I consisted of insecticide-treated plants and group II of untreated samples to allow better visualization of the variation and to present combinations of out/input variables in a specific way. This scrutiny was calculated with PQStat software (ver. 1.6.4). An exploratory method Canonical Variety Analysis (CVA) was applied to cope with collinear high-dimensional data and accordingly grouped variables for yield grain were interpreted. Individual or group mean scores and multiply regression were computed by PaST 3.20 [84]. Multivariate analysis DCA (Detrended correspondence analysis) based on the data matrix revealed the variation patterns in genotypes treated and untreated with insecticide. Variables were computed with Canoco for Windows 4.5. Current work provides data related to multiply regression represented by 8 parameters. GAM (generalized additive model) is defined as the range of models examined step-wise and can be used by adding other covariates. The model was performed with Statistica 13.3. Formally, the GAM framework imposing a particular functional form that may be adapted to the data as a sum of independent variables. *V. unguiculata* has strong potential as a component of a sustainable crop farming system as well as animal feed and fodder. Consequently, our model was determined by a response variable to get an accurate point of forecasting, and was formally written as:(1)$$g\left(E\left(y_{i}\right)\right)=\beta_{0}+f_{1}\left(x_{i1}\right)+\ldots +f_{p}\left(x_{ip}\right)+\varepsilon_{i}$$
where:*g*—link function (the identical function),*y*—response variable (depends linearly on unknown smooth functions),*x*_1_… *x*_*n*_—independent variables (predictor variables),*f*_1_, …, *f*_*p*_—smooth functions (splines),*i* = 1, …, N,*β*—an intercept,*Ɛ*—random error (a constant error variance is assumed).

Interaction between smooth function resulted in the following formula:(2)$$f_{x}=\sum_{i=1}^{q}b_{1}\left(x\right)\beta_{i}$$
where:*f*—smooth functions,*q*—basis dimension,*b*—the sum of basis functions,*β*—corresponding regression coefficients.

## 5. Conclusions

The evaluation of different genotypes of *V. unguiculata* highlighted the adaptability to edaphoclimatic conditions of incorporated genotypes and the interaction between insecticides use and their productivity. Certainly, the use of insecticide contributed to the increase of the yield of *V. unguiculata* seeds. Our findings showed that genotypes IT07K-181-55 (V3), H4 (V7), and IT97K-556-4M (V5) can be recommended to farmers to optimize the yield and integrated management strategy for phytosanitary control of *Zonocerus variegates*. Yield seed was positively correlated with pod number, while seed number per pod and mean seed weight remained negatively related to field potential. Yield may be used specifically as plant identification of adaptability to drought-prone conditions. In future research, *V. unguiculata* should be successfully used as a cover to the subsequent crop and plant protection. Small-scale farmers may provide quantitative traits of field seed for improved cultivar development, large seed size, and resistance to major diseases. Our results showed that seed yield can severely limit the crop system. In the prospect study, we suggest focusing on the selection of suitable genotypes of *V. unguiculata* that dominate in the cultivated areas, including risk assessment of the *Zonocerus variegates*.

## Figures and Tables

**Figure 1 plants-10-01074-f001:**
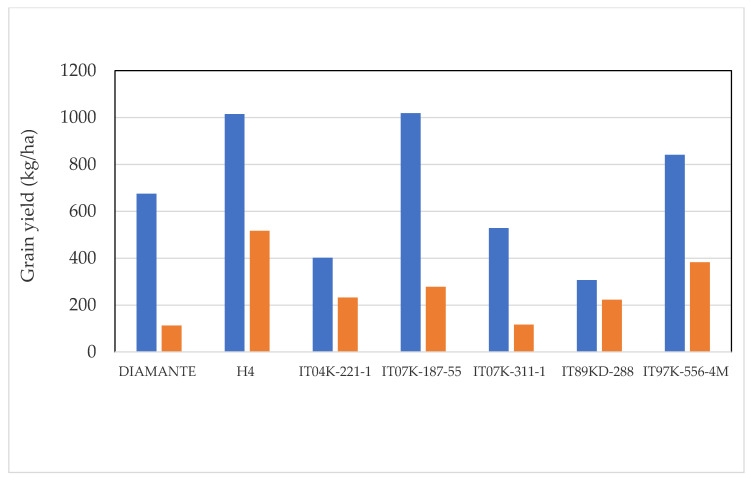
Mean values of grain yield for genotypes treated (blue) and untreated (orange).

**Figure 2 plants-10-01074-f002:**
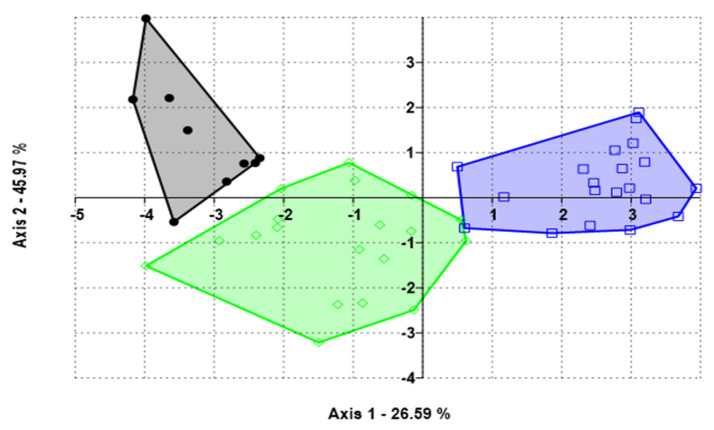
CVA distinguished the best genotypes according to yield (grain). Cycles represent genotype H4, square determine genotype IT07K-181-55 and diamonds indicate genotype IT97K-556-4M. All data is prepared for treated with insecticide.

**Figure 3 plants-10-01074-f003:**
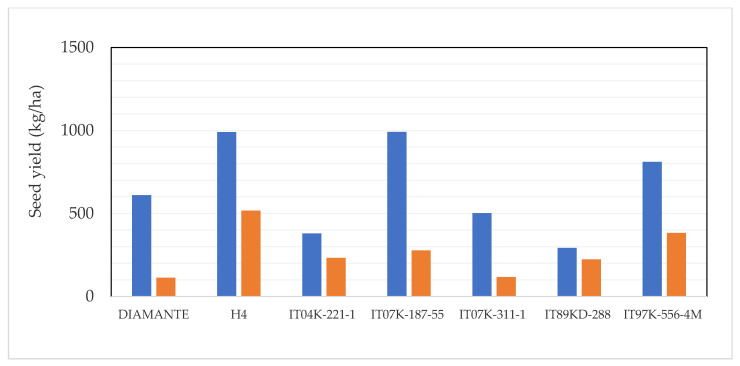
Mean values of seed yield for genotypes treated (blue) and untreated (orange).

**Figure 4 plants-10-01074-f004:**
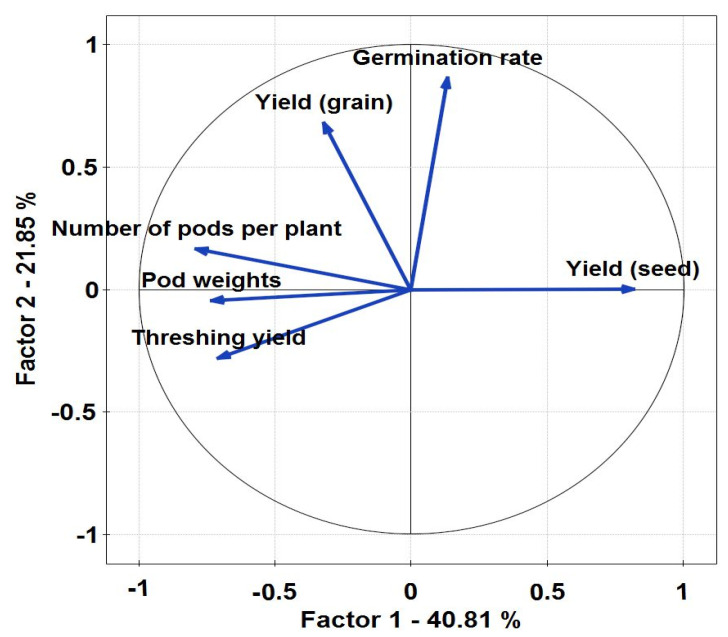
PCA for the studied parameters. The most important factors were germination rate and seed yield. Kaiser-Mayer-Olkin (KMO) coefficient 0.62; *p* < 0.001.

**Figure 5 plants-10-01074-f005:**
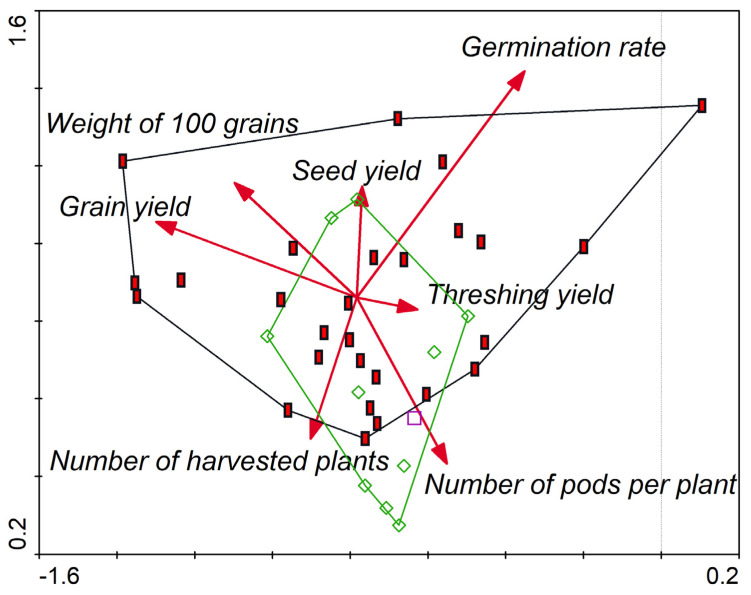
DCA quoted results for studied parameters between treated (red squares) and untreated (green diamonds) plant with insecticide. First axis explained 27.29% of variance and the second axis explained 23.40% of all the total variance.

**Figure 6 plants-10-01074-f006:**
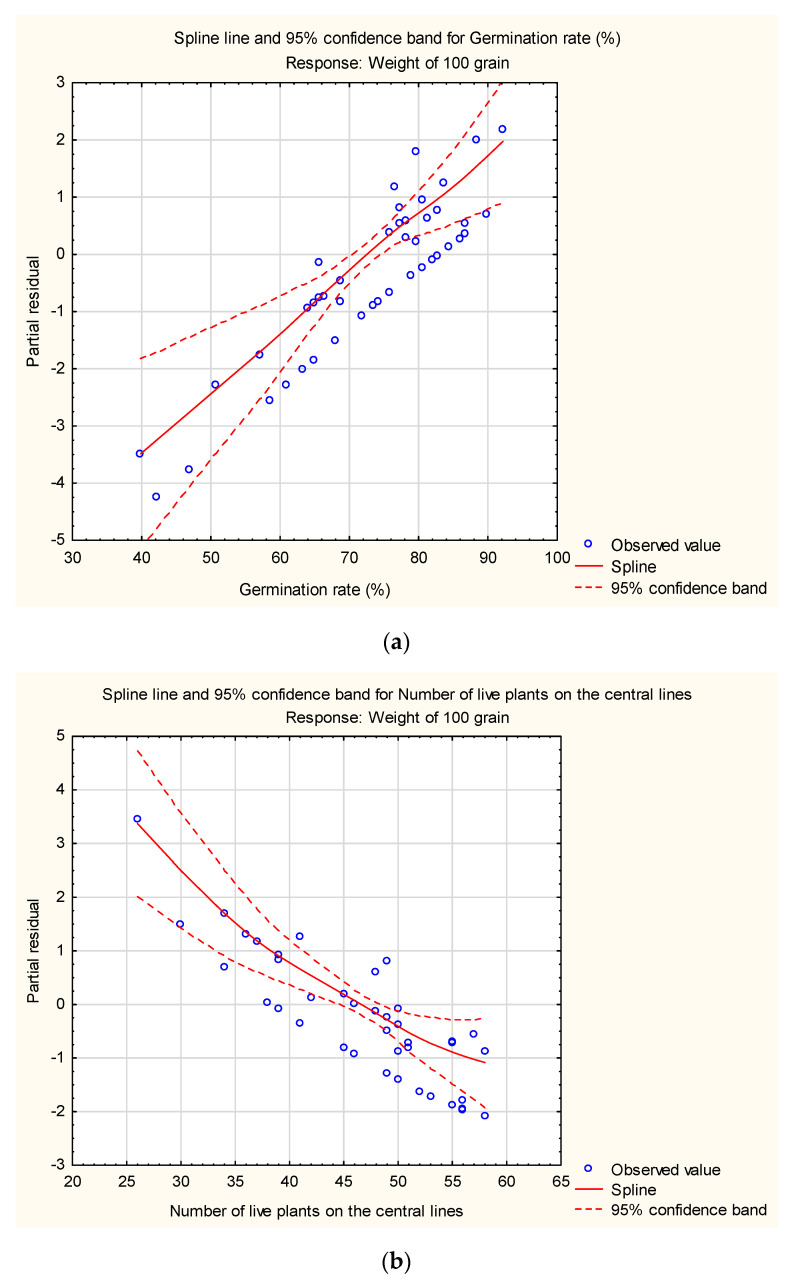
(**a**). GAM for germination rate as covariate and weight of 100 grain as response variable during abundance of *Zonocerus variegates*. (**b**). GAM for number of live plants on the central lines as covariate and weight of 100 grain as response variable during no occurrence of *Zonocerus variegates*. (**c**). GAM for number of harvested plants on the central lines as covariate and weight of 100 grain as response variable. (**d**). GAM for number of live plants in the border lines as covariate and weight of 100 grain as response variable.

**Figure 7 plants-10-01074-f007:**
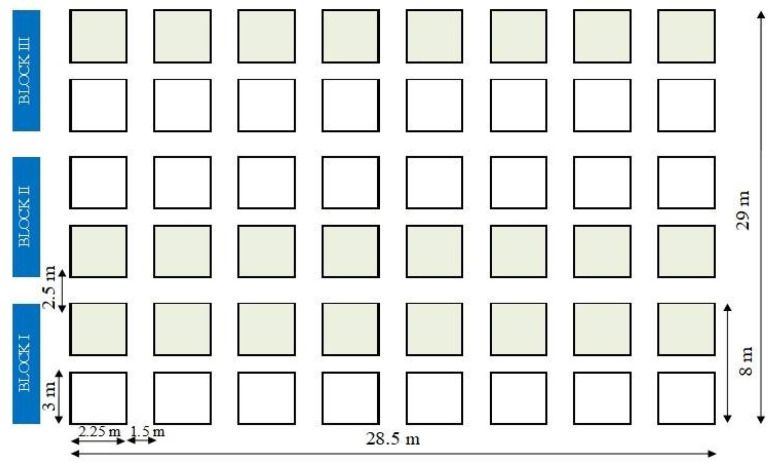
Experimental layout using split-plot design in the study area.

**Table 1 plants-10-01074-t001:** Germination rate for examined genotypes. The average of the genotypes indicated by the letters, are not significantly different at the rate level of 5% and coefficient of variation (CV).

Genotypes	Germination Rate (%)
IT07K-311-1	86 a
IT04K-221-1	78 ab
DIAMANTE	77ab
IT07K-187-55	77 ab
IT89KD-288	75 b
H4	63 c
IT97K-556-4M	53 c
Mean	73
*p* < 0.001; CV = 11%	

**Table 2 plants-10-01074-t002:** Loading factors highlighted by PCA.

Variable	Factor 1	Factor 2	Factor 3	Factor 4	Factor 5	Factor 6
Grain yield	−0.32	0.68	0.55	−0.32	0.13	−0.13
Germination rate	0.13	0.86	−0.27	0.36	0.07	0.17
Seed yield	0.82	0.00	−0.27	−0.04	0.42	−0.28
Number of pods per plant	−0.79	0.17	−0.41	0.08	−0.21	−0.36
Pod weights	−0.73	−0.04	−0.46	−0.35	0.30	0.20
Threshing yield	−0.71	−0.28	0.32	0.43	0.36	−0.04

**Table 3 plants-10-01074-t003:** Eigenvectors computed by PCA.

Variable	Factor 1	Factor 2	Factor 3	Factor 4	Factor 5	Factor 6
Grain yield	−0.20	0.59	0.57	−0.43	0.19	−0.24
Germination rate	0.08	0.75	−0.28	0.49	0.10	0.31
Seed yield	0.52	0.002	−0.28	−0.05	0.61	−0.51
Number of pods per plant	−0.50	0.14	−0.43	0.11	−0.31	−0.66
Pod weights	−0.47	−0.04	−0.48	−0.47	0.45	0.37
Threshing yield	−0.45	−0.24	0.33	0.59	0.53	−0.07

**Table 4 plants-10-01074-t004:** Correlation for the most important parameters subjected by PCA.

Variable	Grain Yield	Germination Rate	Seed Yield	Number of Pods Per Plant	Pod Weights	Threshing Yield
Grain yield	1	0.26	−0.3	0.13	0.08	0.12
Germination rate	0.26	1	0.15	0.1	−0.08	−0.25
Seed yield	−0.3	0.15	1	−0.52	−0.39	−0.52
Number of pods per plant	0.13	0.1	−0.52	1	0.59	0.36
Pod weights	0.08	−0.08	−0.39	0.59	1	0.34
Threshing yield	0.12	−0.25	−0.52	0.36	0.34	1

**Table 5 plants-10-01074-t005:** Result of a generalized additive model for the most essential studied plant traits.

	Variable Index	DF	GAM Coefficient	Standard Error	Standard Score	Non-Linear *p*-Value
Inception	0	1.000000	−1.27733	0.6999	−1.82497	-
Weight of 100 grain	1	4.080600	0.00340	0.0046	0.73912	0.025104
Seed yield	2	4.115461	0.00191	0.0022	0.86949	0.045038
Grain yield	3	4.061803	−0.05003	0.0438	−1.14305	0.102087
Weight of seeds selected	4	3.912032	0.02932	0.0107	2.72839	0.008383
Weight of seeds not selected	5	4.006387	0.05173	0.0265	1.95248	0.005508
Pod weights	6	4.003659	0.10427	0.0253	4.12764	0.874075
Number of harvested plants	7	3.955695	−0.12843	0.0324	−3.96647	0.087135
Germination rate	8	3.975686	−1.29209	569.9148	−0.00227	0.000000
Number of live plants on the central lines	9	4.061591	−0.00001	0.0000	−0.13349	0.000328
Number of live plants in the border lines	10	4.032929	0.00024	0.0004	0.64662	0.000000

**Table 6 plants-10-01074-t006:** Multiply regression for crop parameters defined by four environmental factors.

		Coefficient	Standard Error	t	*p*	R^2^
	Constant	1084.30	2812.90	0.39	0.70	
	Weight of seeds not selected	65.93	50.32	1.31	0.20	0.50
	Number of pods per plant	12.20	319.51	0.04	0.97	0.19
Grain yield	Number of harvested plants	−45.60	55.79	−0.82	0.42	0.05
	maturity (95%)	19.99	37.61	0.53	0.60	0.04
	Pod weights	−7.18	39.44	−0.18	0.86	0.47
	weight of 100 grain	288.94	228.62	1.26	0.22	0.31
	Constant	7.68	0.33	23.23	0.00	
	Weight of seeds not selected	0.01	0.01	1.07	0.29	0.07
	Number of pods per plant	−0.05	0.04	−1.32	0.20	0.11
Risk of aphid-mosaic virus disease	Number of harvested plants	0.01	0.01	1.88	0.07	0.14
	Maturity (95%)	−0.07	0.00	−15.85	0.04	0.88
	Pod weights (gr)	0.00	0.00	−1.06	0.30	0.11
	Weight of 100 grain	−0.06	0.03	−2.19	0.04	0.19
	Constant	−0.46	3.28	−0.14	0.89	
	Weight of seeds not selected	−0.01	0.06	−0.17	0.87	0.80
	Number of pods per plant	0.33	0.37	0.90	0.38	0.40
Weight of selected seeds	Number of harvested plants	0.01	0.07	0.13	0.90	0.16
	maturity (95%)	0.00	0.04	−0.10	0.92	0.05
	Pod weights	0.14	0.05	2.98	0.01	0.85
	weight of 100 grain	0.20	0.27	0.74	0.46	0.40
	Constant	2.15	0.77	2.78	0.01	
	Weight of seeds not selected	0.01	0.01	0.51	0.62	0.24
	Number of pods per plant	−0.10	0.09	−1.11	0.28	0.29
*Zonocerus variegates* prevalence	Number of harvested plants	−0.01	0.02	−0.64	0.53	0.11
	Maturity (95%)	0.03	0.01	3.23	0.35	0.03
	Pod weights	−0.01	0.01	−0.76	0.46	0.23
	weight of 100 grain	−0.17	0.06	−2.71	0.01	0.35

**Table 7 plants-10-01074-t007:** Climate conditions observed from September 2017 to February 2018 throughout the experiment.

Year/2017–2018	Precipitation (mm)	Average Temperature °C	Relative Humidity (%)	Insolation (Calories in cm^3^)
September	36.0	27.5	71.4	590.2
October	214.0	27.4	76.7	622.1
November	206.0	27.4	75.9	671.6
December	234.6	27.4	82.0	568.3
January	99.8	27.9	81.8	731.5
February	94.2	27.1	77.2	641.1

Source: Experimental Station of the Institute of Agricultural Development (IDA) Dange-Quitexe municipality.

## Data Availability

Not applicable.

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
