# Peer review of "Modeling of Cowpea (Vigna unguiculata) Yield and Control Insecticide Exposure in a Semi-Arid Region"

_plants, 2021, doi:10.3390/plants10061074_

Round 1

Reviewer 1 Report

The main problem is that you luck repetition on time and also different similar sites. I think you need to repeat your experimentation at least for one more year.

Also there is no information for which pests and why you need the insecticide.

See some other comments on the attached file

Author Response

Authors:

We express our gratitude for your thoughtful comment and appreciation of our results in the review. This plant has two important qualities. First of all, it has economic value for the region, as it is the staple food. Secondly, it easily adapts to soil conditions and is suitable for growing in difficult climatic conditions. Pest and the need for insecticide application have been incorporated into the text. Please check the Introduction section and the first objective of our study.

Thank you for your comment. Yes, it is a truth that data requires one more year to increase its quality. However, cowpea (Vigna unguiculta) is an important crop in many countries, and especially in African countries. Support for obtaining more productive cultivars and better cultural techniques are important to increase production. Therefore, this work is important for countries like Angola.

Awkward sentences have been removed from the text according to suggestions in the .pdf version.

Reviewer 2 Report

Review

  1. The need of these research conduction is not sufficiently justified in the Introduction . It was not given if pests are the great problem in the cowpea cultivation in region of research or only Zonocerus variegatus (L.) is the most dangerous pest in cowpea plantation.
  2. On the Fig. 2 is lack of T1 and T2 markings.
  3. In the Material and Methods is given that the I row factor is 7 genotypes of cowpea but on the Fig. 7 is 8 genotypes. It should be given that the I row factor was 8 genotypes of cowpea ( in that 7 –new genotypes + 1 local genotype ). Next, it should be explained if local genotype was an experimental object as the new 7 genotypes i.e. if the plots with local genotype were randomly selected in blocks as the other genotypes.
  4. What was the aim of use of local genotype in experiment when later it was not take into consideration at analysis of obtained results. Why the comparison of yield of 7 new genotypes and local genotype were not performed (Fig.1 and 3) ?
  5. Yield from plot with area 6,75m2 should not be counted on tonnes per hectare but it should be given in kg per m2.
  6. In the Material and Methods was 2 fold wrote a sentence „V. unguiculata has strong potential as a component of a sustainable crop farming system as well as animal feed and fodder”.
  7. Which insecticide was used in these researches?
  8. I don’t understand this text „Grain yield (GY) was determined by the number of grains. Seed yield (SY) was determined by the number of selected grains. Weight of 100 grains (WHG) was obtained by the means of the total weight of seeds per treatment. Total yield (TY) was determined as a function of the total productivity of the studied area. since abundance of Z. variegatus in short fallows and adjacent fields was observed”.
  9. There is lack of information on the time of this experiment duration and characteristics of weather conditions. The Authors wrote only that „The lowest annual average precipitation is 100 mm and the highest is 1750 mm”. And how weather was in the years of the research?
  10. There is lack of information how was evaluated „Risk of aphid-mosaic virus disease”.

Author Response

Comments to Authors

  1. The need for these research conduction is not sufficiently justified in the Introduction. It was not given if pests are the great problem in the cowpea cultivation in region of research or only Zonocerus variegatus (L.) is the most dangerous pest in cowpea plantation.

Thank you for a suggestion. We implemented relevant information on cowpea cultivation and the risk of pests in the Introduction section. Please check.

  1. On the Fig. 2 is lack of T1 and T2 markings.

Figure 2 shows only pesticide-treated data. We have included this information in the Figures caption.

  1. In the Material and Methods is given that the I row factor is 7 genotypes of cowpea but on the Fig. 7 is 8 genotypes. It should be given that the I row factor was 8 genotypes of cowpea ( in that 7 –new genotypes + 1 local genotype ). Next, it should be explained if local genotype was an experimental object as the new 7 genotypes i.e. if the plots with local genotype were randomly selected in blocks as the other genotypes.

We express our gratitude for your thoughtful comment. An explanation has been done in the text. Please check chapter 4.2. Sampling method experimental layout and design.

  1. What was the aim of use of local genotype in experiment when later it was not take into consideration at analysis of obtained results. Why the comparison of yield of 7 new genotypes and local genotype were not performed (Fig.1 and 3) ?

Thank you for your comments and great feedback. Initially, we planned to take into account all parameters and genotypes. However, the local variety was slightly susceptible to the pest and we were not able to collect enough data to compare with new genotypes. Relevant information on this issue is provided in the text: “Meanwhile, the local insecticide-treated genotype was able to grow with no signs of pest damage and virus disease. It should be strongly noted that the untreated genotype was susceptible to attacks by Zonocerus variegatus".

  1. Yield from plot with area 6,75m2 should not be counted on tonnes per hectare but it should be given in kg per m2.

We do not know exactly which part of the plot the Reviewer took into account. In our plot, there was no 6.75 m2 size.

  1. In the Material and Methods was 2 fold wrote a sentence „V. unguiculata has strong potential as a component of a sustainable crop farming system as well as animal feed and fodder”.

Thank you for a detailed review. We have removed that sentence.

  1. Which insecticide was used in these researches?

We have applied insecticide with a selective substance – cypermethrin (25%).

  1. I don’t understand this text „Grain yield (GY) was determined by the number of grains. Seed yield (SY) was determined by the number of selected grains. Weight of 100 grains (WHG) was obtained by the means of the total weight of seeds per treatment. Total yield (TY) was determined as a function of the total productivity of the studied area. since abundance of Z. variegatus in short fallows and adjacent fields was observed”.

That issue has explained in the text. We may give more definition, if necessary. An awkward sentence has been deleted.

  1. There is a lack of information on the time of this experiment duration and characteristics of weather conditions. The Authors wrote only that „The lowest annual average precipitation is 100 mm and the highest is 1750 mm”. And how weather was in the years of the research?

Weather data has been put into Table 7.

  1. There is lack of information how was evaluated „Risk of aphid-mosaic virus disease”

The short method has been explained. Please check. The risk of disease has been evaluated on a 1–9 scale (> 7 severity of insect risk). Phytosanitary control was reported according to the rating scale:

1 = no attack

2-3 = slightly

4-6 = moderate

7-9 = severe

Reviewer 3 Report

The numbering of citations in the text must begin with 1. The references should be numbered in order. In References the works must not be in the alphabetical order of the authors, but in the order of citation in the paper.
The explanation of "CVA" must appear when it first appears in the paper.
What is "Grain production" compared to "seed production"?
What is the difference between "seeds selected" and "seeds not selected"?
What is the correct name: Diamonds or Diamond? There are both variants in the paper.

Is "6.338 kg / ha" correct? Or, in English, more correctly like this: 6,338 kg / ha? It also applies to other cases.
wrong “INERA), [44].”, correct INERA) [44].
And here's a typo, right? ”Of the studied area. since abundance of ”
The location of the plots in Figure 7 does not resemble that of Figure 8.
”He genotypes treated (eight) and untreated (eight) with insecticide, using Statistical Software 8.0. In this study, a total of 7 genotypes were characterized. ” Are there 8 genotypes or 7?
When (in what year) did the experience take place?

Author Response

The numbering of citations in the text must begin with 1. The references should be numbered in order. In References the works must not be in the alphabetical order of the authors, but in the order of citation in the paper.
The explanation of "CVA" must appear when it first appears in the paper.

Authors:

Thank you for your attention. CVA was explained when it first appears in the paper.

Reviewer

What is "Grain production" compared to "seed production"?
What is the difference between "seeds selected" and "seeds not selected"?

Authors:
We have explained this issue in the text. We may give a more accurate definition, if necessary.

Reviewer
What is the correct name: Diamonds or Diamond? There are both variants in the paper.

Authors:

Thank you for your very detailed comments. The correct name of the genotype is DIAMANTE. Please be careful! Diamond is also the name of the graph created with CVA. Try to check Figure 2.

Reviewer

Is "6.338 kg / ha" correct? Or, in English, more correctly like this: 6,338 kg / ha? It also applies to other cases.

wrong “INERA), [44].”, correct INERA) [44].

And here's a typo, right? ”Of the studied area. since abundance of ”

Authors:

Thank you for a detailed comment. It should be “6.338 kg / ha”. Awkward sentence has been eliminated.

Reviewer
The location of the plots in Figure 7 does not resemble that of Figure 8.
”He genotypes treated (eight) and untreated (eight) with insecticide, using Statistical Software 8.0. In this study, a total of 7 genotypes were characterized. ” Are there 8 genotypes or 7?

Authors:

Thank you for your comments and great feedback. Initially, we planned to take into account all parameters and genotypes. However, the local variety was not susceptible to the pest and we were not able to collect enough data. Relevant information on this is provided in the text: “Meanwhile, the local insecticide-treated genotype was able to grow with no signs of pest damage and virus disease. It should be strongly noted that the untreated genotype was susceptible to attacks by Zonocerus variegatus".

We consider that all other statements are properly selected and important for the development of research in this field. Overall, it was really well written and interesting comment. We assumed that most of the issues related to this manuscript have been addressed properly.

Reviewer
When (in what year) did the experience take place?

Authors:

Thank you for this comment. Data has been put into Table 7.

Round 2

Reviewer 1 Report

Dear authors you made a good effort to improve your manuscript and considering the nature of the special issue of Plants I think that the paper should be published with minor english spell checks

Author Response

First of all we would like to thank you for your constructive comments and suggestion. We considered the results of this research significant because are suitable to be used by several scientific fields highlighting the agroecology and plant production branch respectively.

The implementation of technical measures and engineering projects requires continued study of these research both in the laboratory and in field conditions. Moreover, we have improved minor English spell and edited text for the second time by expert from Foreign Language Unit at the University of Agriculture in Krakow.
